# Pharmacodynamics, Pharmacokinetics, and Kidney Distribution of Raw and Wine-Steamed Ligustri Lucidi Fructus Extracts in Diabetic Nephropathy Rats

**DOI:** 10.3390/molecules28020791

**Published:** 2023-01-12

**Authors:** Ruqiao Luan, Pan Zhao, Xuelan Zhang, Qinqing Li, Xinfeng Chen, Ling Wang

**Affiliations:** 1Innovative Institute of Chinese Medicine and Pharmacy, Shandong University of Traditional Chinese Medicine, Jinan 250355, China; 2School of Pharmacy, Shandong University of Traditional Chinese Medicine, Jinan 250355, China; 3Shandong Provincial Collaborative Innovation Center for Quality Control and Construction of the Whole Industrial Chain of Traditional Chinese Medicine, Jinan 250355, China; 4School of Pharmacy, Shanxi University of Chinese Medicine, Jinzhong 030619, China

**Keywords:** Ligustri Lucidi Fructus extract, wine-steaming, pharmacodynamics, pharmacokinetics, kidney distribution, diabetic nephropathy rats

## Abstract

The purpose of this study was to investigate differences in the pharmacodynamic, pharmacokinetic, and kidney distribution between Ligustri Lucidi Fructus (LLF) and wine-steamed Ligustri Lucidi Fructus (WLL) extracts in diabetic nephropathy (DN) rats. The DN rats were induced by high-fat-sugar diet (HFSD)/streptozotocin (STZ) regimen. For pharmacodynamics, the DN rats were treated with LLF and WLL extracts to assess the anti-diabetic nephropathy effects. For pharmacokinetics and kidney distribution, the concentrations of drugs (hydroxytyrosol, salidroside, nuezhenidic acid, oleoside-11-methyl ester, specnuezhenide, 1‴-O-β-d-glucosylformoside, G13, and oleonuezhenide) were determined. Regarding the pharmacodynamics, LLF and WLL extracts decreased the levels of blood glucose, serum creatinine (SCr), blood urea nitrogen (BUN), and 24-h urinary protein (24-h Upro) in DN rats. Furthermore, LLF and WLL extracts increased the level of high-density lipoprotein cholesterol (HDL-C); decreased the levels of total cholesterol (TC), triglycerides (TG), low-density lipoprotein cholesterol (LDL-C); and reduced levels of pro-inflammatory cytokines (IL-1β, TNF-α, and IL-6) in DN rats. The anti-diabetic nephropathy effect of the WLL extract was better than that of the LLF extract. Regarding the pharmacokinetic and kidney tissue distribution, there were obvious differences in the eight ingredients between LLF and WLL extracts in DN rats. LLF and WLL extracts had protective effects on DN rats, while the WLL extract was better than the LLF extract regarding anti-diabetic nephropathy effects. The pharmacokinetic parameters and kidney distribution showed that wine-steaming could affect the absorption and distribution of the eight ingredients. The results provided a reasonable basis for the study of the clinical application and processing mechanism of LLF.

## 1. Introduction

Diabetic nephropathy (DN), the main cause of chronic kidney disease and end-stage kidney disease, is a microvascular complication caused by diabetes with a prevalence of 30–40% [1,2]. Traditional Chinese Medicine (TCM) contains a variety of effective components, which can produce comprehensive therapeutic effects through multiple channels and multiple targets, and has obvious advantages and characteristics in the prevention and treatment of chronic metabolic diseases. Moreover, TCM has shown a good effect in the treatment of DN [3,4,5].

Ligustri Lucidi Fructus (LLF) is the dried ripe fruit of *Ligubtrum lucidum* W.T. Aiton [Oleaceae]; it is often used to nourish the liver and kidney, improve eyesight, and blacken hair in clinical practice [6]. Among the clinical prescriptions, LLF is a core drug for the treatment of DN [7]. In the treatment of DN, LLF is often mixed with other TCMs, which can improve the therapeutic effect of TCMs [8,9]. Several pharmacological studies have shown that LLF has anti-inflammatory, antioxidant, antitumor, hepatoprotective, and immune-regulative effects [10,11,12,13,14]. In addition, research has suggested that LLF can also reduce hypercholesterolemia [15]. In terms of the chemical composition, LLF includes phenylethanols, iridoids, triterpenoids, flavonoids, etc [16], of which phenylethanols and iridoids are the main bioactive components. Phenylethanoid components (including salidroside, tyrosinol, hydroxyltyrosinol, acteoside, and echinacoside) have anti-inflammatory, anti-cancer, hypoglycemic, and lipid effects [17,18,19,20]. The iridoid glycosides, which mainly include specnuezhenide, ligustroside G13 (G13), oleonuezhenide, nuezhenidic acid, neonuezhenide, 1‴-O-β-d-glucosylformoside, and oleuropein, were reported for have the effect of lowering blood lipids, anti-oxidation action, and immuno-enhancing activity [21,22,23].

Most TCMs are used clinically after processing to reduce toxicity or enhance the curative effect. According to the Chinese Pharmacopoeia, the processing method of LLF is to soak in rice wine [6]. According to the TCM theory, LLF can strengthen the function of nourishing the liver and kidney after wine-steaming; thus, wine-steamed Ligustri Lucidi Fructus (WLL) is usually used in the clinic. The results showed that the contents of specnuezhenide, G13, oleonuezhenide, neonuezhenide, oleuropein, and nuezhenoside decreased after wine-steaming, while the content of salidroside, tyrosol, and hydroxytyrosol increased [24,25]. According to the pharmacology research, LLF steamed with wine can enhance the protective effects on the oxidative stress damage to the testis of rats with kidney-yin deficiency [26]. It has been reported that the protective effect of WLL on liver injury induced by carbon tetrachloride (CCl_4_) in mice was stronger than LLF [27]. WLL could improve oxidative stress, inhibit the release of inflammatory factors, and regulate endocrine disorders, which can improve and regulate liver and kidney yin deficiency syndrome caused by anger in rats [28]. The studies have indicated that LLF could effectively recover metabolic disorder of blood glucose and blood lipids in DN rats and has excellent prevention and curing effects for DN [29]. However, up to now, the differences in the pharmacodynamics in DN between LLF and WLL extracts have not been researched. 

Several studies have investigated the pharmacokinetics and distribution of active components of LLF in rats [30,31,32,33]. The pharmacokinetic properties of LLF indicated that LLF compounds can be better absorbed by the blood and maintained for a long period of time [30]. Additionally, the UPLC-MS method was used to study the pharmacokinetics of salidroside in control and ovariectomized rats; the results demonstrated that the t1/2, MRT0–∞, and apparent volume of distribution for salidroside increased in ovariectomized rats compared with control rats [31]. Recently, our research team established a UPLC-MS/MS method, which revealed differences in the pharmacokinetic characteristics of nine active components in LLF and WLL in the plasma of normal rats [32]. Zhang et al. studied the distribution of seven active compounds of LLF before and after wine-steaming in the main organs and tissues of rats. The results showed that wine-steaming has an improved effect on the tissue distribution of the active components of LLF in rats [33]. However, a comparison of the pharmacokinetics and distribution of the main active components among raw and wine-processed products of LLF in diseased rats (DN rats) have not been reported.

In this study, a rat model of DN induced by a high-fat-sugar diet (HFSD)/streptozotocin (STZ) was established. Using this animal model, the differences in the protective effects of LLF and WLL on the kidneys of DN rats were tested. Subsequently, the UPLC-MS/MS method was used to determine the eight components in the plasma and kidney of the DN rats. The differences in the pharmacokinetic properties of LLF and WLL in the plasma and the distribution in the kidney were compared in DN rats. This study will provide a basis for clarifying the mechanism of WLL and further study of the clinical efficacy.

## 2. Results

### 2.1. HPLC Method Validation

The HPLC method was validated in terms of calibration curve, precision, stability, repetition, and recovery. This method was selective with no obvious interferences. The calibration curves showed good linearity; the typical equations of the calibration curves and linearity ranges for the eight analytes containing hydroxytyrosol, salidroside, nuezhenidic acid, oleoside-11-methylester, specnuezhenide, 1‴-O-β-d-glucosylformoside, G13, and oleonuezhenide are shown in Table 1. The results showed that there was excellent correlation between the ratio of the peak area and concentration for each component within the linearity ranges. The precision of the eight standard solutions within was lower than 3% RSD. The RSD of the recovery was between 0.41 and 1.44%, the RSD of the repetition test of hydroxytyrosol, salidroside, nuezhenidic acid, oleoside-11-methylester, specnuezhenide, 1‴-O-β-d-glucosylformoside, G13, and oleonuezhenide was 2.46%, 1.36%, 1.74%, 1.73%, 1.65%, 1.28%, 1.79%, and 2.08%, respectively, and RSD of the stability test was 2.46%, 1.14%, 1.28%, 1.68%, 1.94%, 1.85%, 2.38%, and 1.15%, respectively. The eight analytes were stable during the whole experiment. The results showed that this method can be used in the determinations of LLF and WLL.

### 2.2. Detection of Active Ingredients in LLF and WLL Extracts by HPLC

As shown in Figure 1 and Figure 2, and in Table 2, the analytical results showed the contents of hydroxytyrosol, salidroside, nuezhenidic acid, oleoside-11-methylester, specnuezhenide, 1‴-O-β-d-glucosylformoside, G13, and oleonuezhenide in LLF and WLL extracts.

### 2.3. Pharmacodynamic Studies in DN Rats

#### 2.3.1. The Effects of LLF and WLL Extracts on Body Weight, Food Consumption, and Water Intake in DN Rats

Compared with the Control group, the body weight decreased (*p* < 0.01), and food consumption and water intake increased in the Model group (*p* < 0.01), which also indicated that the DN model was made successfully. Compared with the Model group, body weight increased in the LLF, WLL, and MET groups, with significant differences in the WLL and MET groups (*p* < 0.01). Food consumption and water intake in the LLF, WLL, and MET groups decreased significantly (*p* < 0.01 or *p* < 0.05) when compared to the Model group, and degree of decline in the WLL group was more than in the LLF group. The results are shown in Table 3.

#### 2.3.2. The Effects of LLF and WLL Extracts on Blood Glucose Levels in DN Rats

Compared with the Control group, the blood glucose of the Model group increased significantly (*p* < 0.01), which also indicated that the DN model was made successfully. Compared with the Model group, the level of blood glucose decreased in the WLL, LLF, and MET groups. The observed effect was comparable to the results obtained from the MET group. The WLL group had better hypoglycemic ability than the LLF group. The results are shown in Figure 3.

#### 2.3.3. The Effects of LLF and WLL Extracts on SCr, BUN, 24 h urine volume, 24 h Upro, and Kidney Index in DN Rats

Figure 4A–E showed that the SCr, BUN, 24-h urine volume, 24-h Upro levels, and kidney index were significantly higher in the Model group than those in the Control group (*p* < 0.01). This result indicated that the DN model was established successfully. Compared with the Model group, the BUN, 24-h urine volume, 24-h Upro, and kidney index levels decreased significantly in LLF group (*p* < 0.05 or *p* < 0.01), while SCr showed no significant difference. In comparison with the Model group, the SCr, BUN, 24-h urine volume, 24-h Upro, and kidney index levels in the WLL group decreased significantly (*p* < 0.01), and the degree of decline in WLL group was more than LLF group. These results indicated that WLL and LLF extracts had a protective effect on the kidneys of DN rats, and the protective effect of the WLL extract was more significant.

#### 2.3.4. The Effects of LLF and WLL Extracts on Lipid Profiles in DN Rats

The results from the lipid profiles in the serum sre shown in Figure 5. In comparison with the Control group, the levels of TC, TG, and LDL-C significantly increased, and HDL-C was decreased, in the Model group (*p* < 0.01). In the LLF and WLL groups, the levels of TG, TC, and LDL-C were lower when compared with the Model group, whereas the level of HDL-C was higher when compared with the Model group. Moreover, the regulation effect of LLF group was inferior to that of WLL group.

#### 2.3.5. The Effects of LLF and WLL Extracts on Inflammatory Cytokines in DN Rats

As shown in Figure 6, the serum concentrations of IL-1β, TNF-α, and IL-6 were significantly higher in the Model group when compared to the Control group (*p* < 0.01). Whereas treatment with LLF and WLL extracts significantly restored the levels of the IL-1β, TNF-α, and IL-6 in DN rats. Moreover, the WLL group showed a better anti-inflammatory effect than the LLF group regarding the level of IL-1β, TNF-α, and IL-6.

#### 2.3.6. The Effects of LLF and WLL Extracts on Histopathological Changes

Under light microscope observation, the kidney tissue structure of the rats in the Control group was normal and showed no abnormal features (Figure 7A). Rats in the Model group showed obvious kidney damage, such as glomerular atrophy, localized fibrosis, inflammatory cell infiltration, partial degeneration of the renal tubular epithelium, and interstitial congestion (Figure 7B), which also indicated that the DN model was successfully made. The kidney injury of rats in the LLF, WLL, and MET groups recovered to varying degrees (Figure 7C–E). Pathological sections of the kidney showed a small portion of atrophic glomeruli, normal renal tubular epithelium, and interstitial small focal inflammation. This indicated that both the LLF group and the WLL group had restored renal histopathological damage, with the WLL group showing a better effect.

### 2.4. Correlation Analysis

Pearson correlation analysis was used to explore the potential relationship between eight active ingredients and the physicochemical parameters after oral administration of LLF and WLL extracts in DN rats (Figure 8). The results shown that all the active components were positively associated with body weight and negatively correlated with water intake, 24-h urine volume, 24-h Upro, TC, and LDL-C. Except for oleoside-11-methylester, G13, and oleonuezhenide, all active ingredients were negatively correlated with food consumption, SCr, BUN, kidney index, TG, IL-1β, TNF-α, and IL-6. In addition, active components such as hydroxytyrosol, salidroside, and nuezhenidic acid were negatively correlated with serum glucose, while other active components were positively correlated with serum glucose. Furthermore, some active components (e.g., hydroxytyrosol, salidroside, nuezhenidic acid, and specnuezhenide) also had a positive correlation with HDL-C. Moreover, it can be clearly seen that hydroxytyrosol, salidroside, and nuezhenidic acid were strongly correlated with the physicochemical parameters (R > 0.85 or R < −0.75), indicating that these components have kidney protective effects.

### 2.5. Plasma Pharmacokinetics in DN Rats

#### 2.5.1. LC-MS/MS Method Validation

The plasma drug determination method has been verified in terms of specificity, calibration curve, sensitivity, accuracy, precision, and stability. This method was selective, has no obvious interference, and has no matrix effect. The calibration curve equations of the eight components are shown in Table 4. The calibration curves showed good linearity and all of the correlation coefficients were higher than 0.9991. The results showed that the ratio of the peak area to the concentration of each compound has a good correlation within the linear range. The RSD expressing the intra-day and the inter-day accuracy of salidroside, hydroxytyrosol, nuezhenidic acid, oleoside-11-methyl ester, 1‴-O-β-d-glucosylformoside, specnuezhenide, G13, and oleonuezhenide in rat plasma at high, middle, and low concentrations were 0.47–10.28% and 0.37–10.08%. The RE expressing the intra-day and the inter-day precision of salidroside, hydroxytyrosol, nuezhenidic acid, oleoside-11-methyl ester, 1‴-O-β-d-glucosylformoside, specnuezhenide, G13, and oleonuezhenide in rat plasma at high, middle, and low concentrations were −3.65–3.52% and −3.39–6.02%.

#### 2.5.2. Plasma Pharmacokinetic Studies in DN Rats

The pharmacokinetic parameters and the mean pharmacokinetic profiles of the eight ingredients of LLF and WLL in DN rats are displayed in Table 5 and Figure 9. Most of the eight ingredients reached the maximum of drug concentration within 1 h after oral administration, indicating that these ingredients exhibited rapid absorption in DN rats. In comparison to LLF, the AUC_0–12 h_ and C_max_ values of salidroside (2527.81 ± 489.80 ng/mL·h, 697.01 ± 131.21 ng/mL), hydroxytyrosol (10,902.26 ± 338.71 ng/mL·h, 3462.78 ± 108.62 ng/mL), and nuezhenidic acid (1615.14 ± 53.47 ng/mL·h, 766.20 ± 62.39 ng/mL) were increased significantly (*p* < 0.05) from the WLL, indicating that the bioavailability of hydroxytyrosol, salidroside, and nuezhenidic acid in WLL was higher than that in LLF. The main reason may be the increase in the content of salidroside, hydroxytyrosol, and nuezhenidic acid after wine-processing.

WLL is the product of LLF after wine-steaming. Wine had the function of promoting blood circulation in the TCM theory [34,35]. After intragastric administration of LLF, the AUC_0–12 h_ and C_max_ values were as follows: specnuezhenide (774.79 ± 38.60 ng/mL·h, 698.99 ± 94.38 ng/mL), oleoside-11-methyl ester (2537.42 ± 67.83 ng/mL·h, 744.81 ± 9.29 ng/mL), 1‴-O-β-d-glucosylformoside (901.37 ± 92.32 ng/mL·h, 655.39 ± 42.24 ng/mL), G13 (1025.82 ± 77.41 ng/mL·h, 500.50 ± 20.10 ng/mL), and oleonuezhenide (605.12 ± 26.46 ng/mL·h, 300.54 ± 27.04 ng/mL). After rats were given WLL by gavage, the AUC_0–12 h_ and C_max_ values were decreased as follows: specnuezhenide (348.81 ± 7.3 ng/mL·h, 195.09 ± 6.81 ng/mL), oleoside-11-methyl ester (2021.07 ± 99.67 ng/mL·h, 605.62 ± 14.86 ng/mL), 1‴-O-β-d-glucosylformoside (531.78 ± 54.16 ng/mL·h, 384.75 ± 38.68 ng/mL), G13 (881.95 ± 39.80 ng/mL·h, 377.69 ± 16.56 ng /mL), and oleonuezhenide (437.77 ± 9.87 ng/mL·h, 223.92 ± 37.50 ng/mL). The contents of specnuezhenide, 1‴-O-β-d-glucosylformoside, oleoside-11-methyl ester, G13, and oleonuezhenide in LLF were 2.36 times, 2.69 times, 10.88 times, 10.77 times, and 23.74 times higher than those in WLL, respectively. The decreasing trend in their content is inconsistent with the decreasing trend in the AUC_0–12h_ and C_max_. Compared with the decreasing trend in their content, the decreasing trend in the AUC_0–12h_ and C_max_ was reduced in WLL, which shows that wine-steaming can promote the bioavailability of specnuezhenide, 1‴-O-β-d-glucosylformoside, oleoside-11-methyl ester, G13, and oleonuezhenide in DN rats.

Based on the above data analysis, it can be shown that wine-steaming can promote the absorption of phenylethanol and other small molecular substances in LLF and promote the absorption of substances to quickly reach the peak blood drug concentration. According to the previous experimental results, specnuezhenide, 1‴-O-β-d-glucosylformoside, G13, and oleonuezhenide can be transformed into salidroside, hydroxytyrosol, and nuezhenidic acid during wine-steaming, which shows that the transformation of LLF components during wine-steaming can lead to the increase in the salidroside, hydroxytyrosol, and nuezhenidic acid content and plasma concentration. The increase of AUC_0–12h_ and C_max_ of these three components also indicates that the bioavailability increased after wine-steaming, and that macromolecular substances are more easily absorbed after conversion to small molecular substances.

### 2.6. Tissue Distribution in DN Rats

#### 2.6.1. Method Validation

The method of determination of kidney drug distribution was validated in terms of specificity, calibration curve, sensitivity, accuracy, precision, and stability. The calibration curve equations and linear ranges of the eight analytes are shown in the Table 6. The standard curve showed a good linear relationship. The RSD expressing intra-day and the inter-day accuracy of salidroside, hydroxytyrosol, nuezhenidic acid, oleoside-11-methyl ester, 1‴-O-β-d-glucosylformoside, specnuezhenide, G13, and oleonuezhenide in rat plasma at high, middle, and low concentrations were 0.72–12.79% and 1.19–12.23%. The RE expressing the intra-day and the inter-day precision of salidroside, hydroxytyrosol, nuezhenidic acid, oleoside-11-methyl ester, 1‴-O-β-d-glucosylformoside, specnuezhenide, G13, and oleonuezhenide in rat plasma at high, middle, and low concentrations were −10.04–13.34% and −11.11–11.68%. The above results show that this method was selective, without obvious interference, and without matrix effect, which can be used for the determination of LLF and WLL in rat kidney tissue.

#### 2.6.2. Tissue Distribution

The mean content of the eight components in the kidney tissue are illustrated in Figure 10. In addition, the content of the eight components in the kidney tissue were calculated and presented in Table 7. After the extracts of LLF and WLL were administered to rats by intragastric administration, the eight components were rapidly distributed in rats. A higher concentration of the drug could be detected in vivo 30 min after administration, and the concentration of the components in kidney decreased significantly 4 h after administration. After wine-steaming, the concentrations of salidroside, hydroxytyrosol, and nuezhenidic acid in the kidney tissues of DN rats increased, while oleoside-11-methyl ester, 1‴-O-β-d-glucosylformoside, specnuezhenide, G13, and oleonuezhenide showed a decreasing trend in concentration in the rat kidney tissue.

## 3. Discussion

DN is a chronic kidney disease, which is mainly characterized by renal capillary rupture, and glomerular capillary and renal tubular interstitial damage [36]. The combination of a HFSD diet combined with STZ injection can induce insulin resistance and significantly increase the levels of blood glucose, which in turn triggers DN [37,38]. This model is very similar to type 2 diabetes in human [39]. Here, insulin resistance was induced by administration of HFSD in rats for 4 weeks, followed by STZ peritoneal injection to establish a successful DN rat model, which is similar to that reported in a previous study. High blood glucose can cause damage to some important tissues and organs in the body [40], producing loss of renal function associated with the elevation of TC, TG, LDL-C, Scr, BUN, and microalbumin levels [41,42]. We observed a significant increase in blood glucose levels in the DN rats. However, 8 weeks of treatment with LLF and WLL extracts gradually reduced the level of blood glucose, with the WLL extract showing a better effect than the LLF extract. In addition, DN rats experienced an increase in water intake and food consumption, and a decrease in body weight due to prolonged hyperglycemia. Surprisingly, treatment with LLF and WLL extracts produced body weight gain, and water intake and food consumption decreased. Furthermore, the kidney index of DN rats has increased due to kidney hypertrophy. Significant changes in parameters related to kidney structure and function were observed after administration, revealing the protective effect of WLL and LLF extracts on kidney damage caused by DN. All structural and functional abnormalities were restored following administration of LLF and WLL extracts; the effect of the WLL extract was better than that of the LLF extract.

SCr and BUN are considered as clinical indicators of DN [43]. High levels of SCr and BUN were found in DN rats, while LLF and WLL extracts reduced these levels. Thus, LLF and WLL extracts exhibited a dramatic protective effect against DN; the effect of the WLL extract was better than that of the LLF extract. Tubular and glomerular injury are often accompanied by high levels of urinary protein [44]. The 24-h Upro of DN rats increased significantly. After treatment with LLF and WLL extracts, the urine excretion of these proteins was significantly reduced; the effect of the WLL extract was better than that of the LLF extract.

Hyperlipidemia is an important cause of kidney damage. Kidney disease is closely related to high levels of TG, TC, and LDL-C, and a low level of HDL-C [41]. In this study, we found that LLF and WLL extracts could restore high LDL-C and low TG, TC, and HDL-C levels to different degrees, indicating that the protective role against DN of WLL was better than that of LLF.

Cytokines are important parameters to identify kidney injury, and several proinflammatory mediators such as TNF-α, IL-6, and IL-1β are closely related to the progression of DN [45]. Oral administration of LLF and WLL extracts can significantly inhibit the release of pro-inflammatory mediators (TNFα, IL-6, and IL-1β) in the serum of DN rats, indicating that the anti-inflammatory effects of LLF and WLL extracts may be related to kidney protection in DN; meanwhile, the anti-inflammatory effect of the WLL extract was better than that of the LLF extract.

LLF mainly contains iridoids and phenylethanoids, which have good pharmacodynamic activity. Our previous experiments showed that the ethyl acetate and n-butanol components of LLF had protective effects in DN rats. The results of the HPLC demonstrated the effectiveness of hydroxytyrosol, salidroside, nuezhenidic acid, oleoside-11-methylester, specnuezhenide, 1‴-O-β-D-glucosylformoside, G13, and oleonuezhenide, which have a high content [46]. We speculated that these components might be the effective material basis in the treatment of DN. It has been reported that salidroside has a protective effect on the kidney of DN rats. Salidroside, tyrosol, and hydroxytyrosol have hypoglycemic, anti-inflammatory, and hypolipidemic effects [47,48]. Specnuezhenide and salidroside are characteristic components of WLL, which have been used as content determination indexes for quality control of Pharmacopoeia 2020 [6]. Therefore, we studied the pharmacokinetics and kidney distribution of these eight components in LLF and WLL extracts in DN rats. Firstly, a UPLC-MS/MS method was used for rapid and accurate quantification of the eight major bioactive components in the plasma of DN rats. Then, the main pharmacokinetic parameters and kidney distribution of the eight main active components in LLF and WLL extracts were analyzed. Compared with LLF, the AUC_0–12 h_ and C_max_ values of salidroside, hydroxytyrosol and nuezhenidic acid increased, while oleoside-11-methyl ester, 1‴-O-β-d-glucosylformoside, specnuezhenide, G13, and oleonuezhenide were decreased in WLL. In kidney tissue, the concentration of salidroside, hydroxytyrosol, and nuezhenidic acid increased in WLL compared with LLF, while oleoside-11-methyl ester, 1‴-O-β-d-glucosylformoside, specnuezhenide, G13, and oleonuezhenide decreased in WLL compared with LLF. The pharmacokinetic characteristic parameters of the AUC_0–12h_ and C_max_ of LLF were significantly different from those of WLL, indicating that steaming with wine could improve the bioavailability of LLF. Our previous research showed that the macromolecule iridoid glycosides are transformed into small molecule phenylethanol components, which are more easily absorbed by the body [49]. The pharmacokinetic study also showed that salidroside had a higher bioavailability and was more conducive to body absorption than specnuezhenide and G13 [50]. In our experiment, the WLL had an enhanced kidney protective effect in DN. It is speculated that the enhancement of kidney efficacy of LLF after wine-steaming may be related to the conversion of macromolecule iridoid glycosides into small molecular phenylethanol components, which is more conducive to absorption.

In addition, it was found that after wine-steaming, the contents of oleoside-11-methyl ester, 1‴-O-β-d-glucosylformoside, specnuezhenide, G13, and oleonuezhenide decreased, but the decrease in the AUC_0–12h_ and C_max_ in DN rats was much less than the contents in DN rats. The same trend was seen in the distribution of kidney tissue. It is speculated that wine-steaming promoted the absorption of these components and increased the bioavailability of these components in DN rats. The results showed that the effect of WLL was enhanced not only because the macromolecular substances were transformed into easily absorbed small molecules, but also because the wine-steaming promoted the absorption of substances, which further proves the theory of the effect of LLF wine-steaming.

## 4. Materials and Methods

### 4.1. Chemicals and Reagents

Crude products of LLF were purchased from Jianlian Shengjia TCM Co. Ltd. (Shandong Province, Jinan, China), which were identified by Professor Li Feng of TCM identification at Shandong University of TCM. Voucher specimen Nos. SDCM-YZ2019040601 were stored in the School of Chinese Materia, Shandong University of TCM (Jinan, China). The specimens were deposited at the laboratory of the author. Acetonitrile, methanol, and formic acid were provided by Fisher Scientific (Waltham, MA, USA). Purified water was purchased from watsons (Jinan, China). Other reagents were of analytical grade. Serum creatinine (SCr), blood urea nitrogen (BUN), 24-h urinary albumin, high-density lipoprotein cholesterol (HDL-C), low-density lipoprotein cholesterin (LDL-C), triglyceride (TG), and total serum cholesterol (TC) assay kits were purchased from Nanjing Jiancheng Bioengineering Institute. ELISA kits for TNF-α, IL-6, and IL-1β were purchased from Wuhan Genmei Biotechnology Co, Ltd. Streptozocin (STZ) was purchased from Sigma Chemicals Co. (St. Louis, MO, USA). Metformin (MET) was purchased from Shanghai Squibb Pharmaceutical Co. Ltd. (Shanghai, China). Metformin was ground into a powder with a mortar before dissolution.

The standards for G13, oleonuezhenide, nuezhenidic acid, and oleoside-11-methyl ester were purchased from Shanghai Yilin Biotechnology Co., Ltd. (Shanghai, China), specnuezhenide was purchased from Chengdu Biopunify Phytochemicals Co., Ltd. (Chengdu, China), hydroxytyrosol and geniposide were provided by Chengdu Must Biotechnology Co., Ltd. (Chengdu, China), and 1‴-O-β-D-glucosylformoside and salidroside were obtained from Shanghai Yuanye Biotechnology Co., Ltd. (Shanghai, China). The purity of these standards was greater than 98%. The chemical structures are presented in Figure 11.

### 4.2. Instruments

The HPLC system consisted of a 1430 diode array detector, 1110 pumps with a high-pressure mixer, a 1310 column oven, and 1210 autosampler (Hitachi, Tokyo, Japan). The chromatographic separation of the analyte was achieved by a Kromasil C18 column (5 μm, 250 mm × 4.6 mm, Akzo Nobel N.V, Amsterdam, The Netherlands).

The UPLC system (Agilent 1260 series; Santa Clara, CA, USA) was equipped with a solvent degasser, G1311B quaternary pump, and G1329B automated injector. A Halo^®^ C18 column (2.1 × 100 mm, 2.1 μm) was used for elution. An Agilent 6470 triple quadrupole mass spectrometer (Santa Clara, CA, USA) equipped with ESI source was used for mass detection.

### 4.3. Preparation of WLL

WLL was prepared in the laboratory. LLF was added with appropriate amount of rice wine and water, soaked for 4 h until soft, placed in a closed container, steamed for 24 h until the surface turned black, and then dried in a 50 °C drying oven.

### 4.4. Preparation of LLF and WLL Extracts

LLF was powdered by pulverizer, accurately weighed to 1.0 kg, and underwent ultrasonic extraction with 50% ethanol three times for 1 h each time; the ratios of plant/ethanol used were 1/10 (*w/w*). The filtrate was combined and ethanol was recovered. The total crude extract of LLF was purified by macroporous resin to obtain the LLF extract, which was accurately weighed. The preparation of WLL extract was the same as the above procedure. The two samples obtained were used in subsequent experiments.

### 4.5. Quantitative Analysis of LLF and WLL Extracts by HPLC

Quantitative determination of the eight ingredients in LLF and WLL extracts were performed. Before analysis, the LLF and WLL extracts (0.10 g) were dissolved in 50 mL of 50% methanol and filtered through a 0.22 μm membrane.

The chromatographic conditions were established by our team in the early stage [51]. The Kromasil C18 column (250 mm × 4.6 mm i.d., 5 μm particle size) was used, the mobile phase was acetonitrile (solution A) and 0.1% formic acid aqueous solution (solution B). The flow rate was 1 mL/min, the column temperature was 20 °C, and the injection volume was set to 10 μL. The sequence of gradient elution was as follows: 0–10 min with 7–12% A; 10–35 min with 12–25% A; 35–52 min with 25–41% A. The wavelength of the UV detector was fixed at 280 nm (hydroxytyrosol and salidroside) and 240 nm (nuezhenidic acid, oleoside-11-methyl ester, specnuezhenide, 1‴-O-β-d-glucosylformoside, G13, and oleonuezhenide) for determination.

### 4.6. Experimental Animals

Five-week-old male Sprague-Dawley (SD) rats, weighing 150 ± 20 g, were purchased from the Jinan Pengyue experimental animal breeding co. Ltd. (Shangdong, Jinan, China). Rats were reared in a 12 h light/dark cycle at 25–30 °C with free access to drinking water and food. Before the experiment, the animals were bred adaptively for 1 week. All animal experiments were performed in accordance with the guidelines established by the Institutional Animal Protection and Use Committee of Shandong University of TCM.

### 4.7. Establishments of DN Rat Models

A high-fat-sugar diet (HFSD)/streptozotocin (STZ) regimen was used to experimentally induce DN in rats. Before the experiment, the rats were adaptively reared for 1 week. The rats in the control group were maintained on a standard diet, and the model rats were fed a HFSD for 4 weeks. The HFSD was composed of 18% fat, 20% sucrose, 3% cholesterol, and 59% basic food. After 4 weeks, the rats were fasted for 12 h. Then, the rats in the Model groups were intraperitoneally injected STZ at 50 mg/kg dissolved in 0.05 mol/L citrate buffer (pH = 4.5). The rats in the Control group received the same volume of citrate buffer as the body weight standard. The blood glucose level of the rats was measured 3 days after STZ injection. Rats with blood glucose levels over 16.6 mmol/L were used as the follow-up experimental model rats.

### 4.8. Pharmacodynamic Study of LLF and WLL Extracts in DN Rats

#### 4.8.1. Experimental Procedure

DN rats were randomly divided into the Model group, LLF group, WLL group, and MET group, with 10 rats in each group. Each group was treated as follows: Control group and Model group were treated with an equal volume of distilled water. The LLF group was given 2.12 g/kg extract of LLF and the WLL group was given 1.89 g/kg extract of WLL, both of which were equivalent to 15 g/kg raw LLF. The MET group was treated with 200 mg/kg metformin. All rats were orally administered once every morning. Blood was collected from the tail vein every 2 weeks to measure blood glucose levels. After the last treatment, the food consumption, water consumption, and body weight of the rats were measured. The rats were placed in a metabolic cage to collect urine samples for 24 h. The collected urine samples were centrifuged at 3000 rpm for 5 min, and the supernatant was taken and stored at −20 °C until use. Blood samples were collected through the abdominal aorta after the rats were anesthetized with isoflurane; samples were centrifuged at 3000 rpm for 10 min to separate the serum and stored at −80 °C for biochemical analysis of kidney function. Finally, the two kidneys of each rat were collected and weighed to calculate the kidney/body mass index. A kidney was taken for histopathological evaluation.

#### 4.8.2. Urine Analysis and Measurement of Biochemical Parameters

24-h urinary protein was measured using the auto spectrophotometer (Shimadzu, Kyoto, Japan). The levels of SCr and BUN were measured using the Spectra Max M 5 automatic microplate reader (Molecular Devices, Inc., San Jose, CA, USA). 

#### 4.8.3. Analysis of Lipid Profiles

The effect of LLF and WLL extracts on the lipid profiles was measured in serum using automatic biochemical analyzer, according to the manufacturer’s instructions. 

#### 4.8.4. Assessment of Inflammatory Cytokines

We investigated the inflammatory cytokines responsible for DN, including TNF-α, interleukin 1 β (IL-1β), and interleukin 6 (IL-6). The concentrations of inflammatory cytokines were measured using ELISA assay kits, according to the manufacturer’s instructions. 

#### 4.8.5. Histopathological Examination

Kidney tissue was collected and stored in 10% paraformaldehyde immediately. Paraffin sections of the kidney were used for hematoxylin and eosin (H&E) staining to observe pathological changes of the kidney tissue.

### 4.9. Plasma Pharmacokinetics in Rats

#### 4.9.1. UPLC-MS/MS Conditions

The mobile phase consisted of 0.1% (*v/v)* formic acid-water (solvent A) and acetonitrile (solvent B). Gradient elution was carried out according to the following program: 15–25% B (0–3 min), 25–75% B (3–4 min), 75% B (4–6 min), 75–15% B (6–7 min), and 15% B (7–9 min). The column temperature was set at 35 °C, the flow rate was 0.3 mL/min, and the injection volume was 5 μL. The mass spectrometer was set at negative ionization mode. The multiple reaction monitoring (MRM) mass scan mode was implemented. The optimized parameters were programmed as follows: nebulizer gas pressure, 50 psi; capillary voltage, 3500 V; cell acceleration voltage, 4 V; gas temperature, 350 °C; dwell time, 30 ms, and flow rate, 11 L/min. The precursor ion/product ion were *m/z* 229.1/199.2 for salidroside, 153.1/123.1 for hydroxytyrosol, 433.2/209.1 for nuezhenidic acid, 403.1/223.1 for oleoside-11-methyl ester, 685.2/299.0 for 1‴-O-β-D-glucosylformoside, 685.5/453.3 for specnuezhenide, 1071.3/771.5 for G13, 1071.4/909.3 for oleonuezhenide, and 411.1/217.0 for the internal standard (IS, geniposide).

#### 4.9.2. Preparation of Standard Solutions, Calibration Samples, and Quality Control Samples

An appropriate amount of each standard solution was dissolved in methanol to prepare a mixture containing salidroside, hydroxytyrosol, nuezhenidic acid, oleoside-11-methyl ester, 1‴-O-β-d-glucosylformoside, specnuezhenide, G13, and oleonuezhenide stock standard solutions. The mixed stock solution was serially diluted with methanol to obtain a series of working standard solutions. The internal standard stock standard solution of geniposide in methanol was similarly diluted to produce a working solution of 600 ng/mL. The diluted stock solution was added to blank rat plasma to prepare the calibration curve working solution. Quality control samples of the eight markers were prepared from blank plasma with high, medium, and low concentrations.

#### 4.9.3. Plasma Sample Treatments

One-hundred microliters of plasma sample was placed in a 1.5 mL EP tube and 10 μL of internal standard working solution (600 ng/mL geniposide) and 300 μL of methanol were added. 

The mixture was vortexed for 3 min, centrifuged for 10 min, and the supernatant was taken and dried with nitrogen; the residue was reconstituted by adding 100 µL of the initial mobile phase. Sixty microliters of the prepared sample was transferred to a liquid phase vial, and 5 μL of the prepared sample underwent UHPLC-MS/MS analysis.

#### 4.9.4. Plasma Sample Analysis

Before the pharmacokinetic study, the DN rats were fasted for 12 h, during which time they had free access to drinking water. Twelve rats were randomly divided into two groups, with six rats in each group. Rats were orally administered LLF and WLL extracts with a dose equivalent to 15 g/kg raw LLF. At 0, 0.083, 0.167, 0.333, 0.50, 0.75, 1, 2, 4, 6, 8, 10, and 12 h after administration, blood samples (500 μL) were collected from the rat orbit and placed in a 1.5 mL heparin sodium anticoagulant centrifuge tube. The sample was immediately centrifuged at 4000 rpm for 10 min and the plasma was stored at −20 °C for subsequent analysis. Data analysis was performed using the DAS 2.0 pharmacokinetic software (Shanghai, China).

### 4.10. Kidney Distribution in Rats

#### 4.10.1. Kidney Tissue Collection

DN rats were randomly divided into two groups: the LLF extract group and WLL extract group. The two groups were orally given the same dose of extracts equivalent to raw LLF. Tissues of kidneys were taken at 0.5, 1, 2, and 4 h after oral administration. The blood and contents on the tissue surface were washed with 0.9% saline solution, dried, and stored at −20 °C before analysis. Six batches were determined in parallel.

#### 4.10.2. Kidney Sample Pretreatment

Four-hundred micrograms of kidney tissue was weighed, cut into small pieces, and normal saline was added at a ratio of 1:2 (*w*/*v*). Tissue homogenate was prepared under ice bath conditions, centrifuged at 12,000 rpm for 5 min, and 100 µL supernatant was taken; 10 µL internal standard solution and 10 µL methanol solution were added, rotated, and mixed for 3 min. Next, 300 µL of methanol solution was added, vortexed for 3 min, centrifuged at 10,000 rpm for 10 min, followed by removal of the supernatant and drying with nitrogen. Then, 100 µL of the initial mobile phase was added to the residue, vortexed and mixed for 3 min, and 5 µL of sample was taken for analysis.

### 4.11. Statistical Analyses

All data were expressed as mean ± standard deviation (SD). Analysis was performed with SPSS software 21.0 (IBM, New York, NY, USA). The statistical significance of the pharmacodynamics, pharmacokinetics, and tissue distribution parameters obtained from each group were evaluated. The data were evaluated using one-way analysis of variance (ANOVA). *p* < 0.05 was considered statistically significant, and *p* < 0.01 was considered very significant.

## 5. Conclusions

In conclusion, both LLF and WLL extracts had protective effects in HFSD/STZ induced DN rats. The renal protective effect of the WLL was better than that of the LLF. Regarding the pharmacokinetics and kidney distribution, wine-steaming could affect the absorption and distribution of salidroside, hydroxytyrosol, nuezhenidic acid, oleoside-11-methyl ester, 1‴-O-β-d-glucosylformoside, specnuezhenide, G13 and oleonuezhenide to different degrees and promoted the absorption of salidroside, hydroxytyrosol, and nuezhenidic acid, which may be the reason for the increased effect of LLF after wine-steaming. In addition, this is the first time that the mechanism of efficacy enhancement after wine-steaming has been revealed from the perspective of the overall pharmacokinetics and kidney distribution. This study provides an important basis for the pharmacological research and clinical application of LLF.

## Figures and Tables

**Figure 1 molecules-28-00791-f001:**
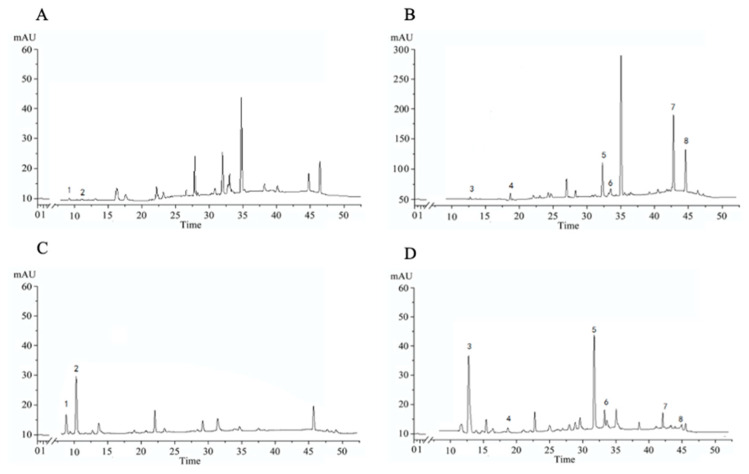
HPLC chromatograms of (**A**) LLF in 280 nm; (**B**) LLF in 240 nm; (**C**) WLL in 280 nm; (**D**) WLL in 240 nm. 1: hydroxytyrosol; 2: salidroside; 3: nuezhenidic acid; 4: oleoside-11-methyl ester; 5: specnuezhenide; 6: 1‴-O-β-d-glucosylformoside; 7: G13; 8: oleonuezhenide.

**Figure 2 molecules-28-00791-f002:**
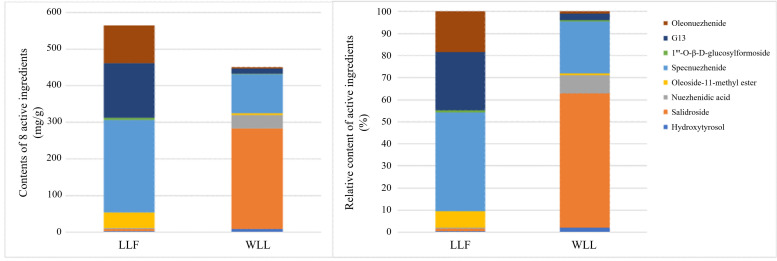
The contents and proportion of the eight active ingredients in LLF and WLL extracts.

**Figure 3 molecules-28-00791-f003:**
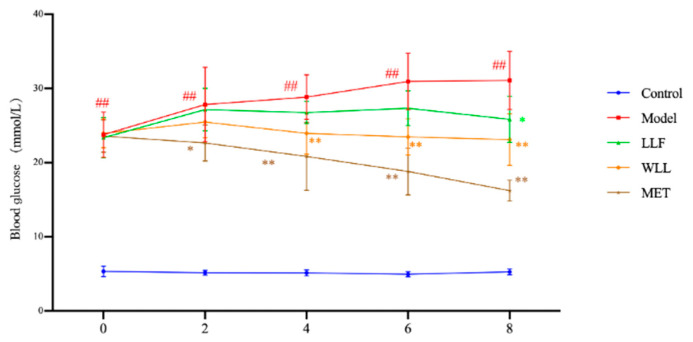
Effects of LLF and WLL extracts on blood glucose levels in DN rats. Significant differences with the Control group were designated as ^##^
*p* < 0.01. Significant differences with the Model group were designated as * *p* < 0.05, ** *p* < 0.01.

**Figure 4 molecules-28-00791-f004:**
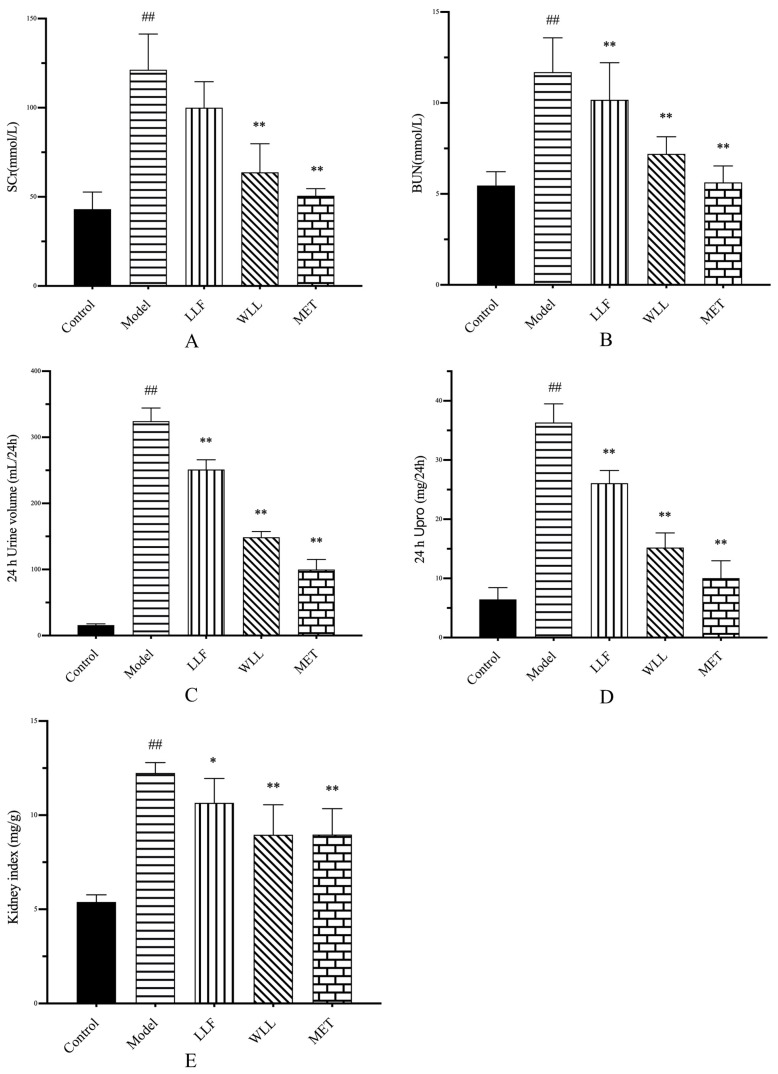
Effects of LLF and WLL extracts on SCr (**A**), BUN (**B**), 24-h urine volume (**C**), 24-h Upro (**D**), and kidney index (**E**) in DN rats. Significant differences with the Control group were designated as ^##^
*p* < 0.01. Significant differences with the Model group were designated as * *p* < 0.05, ** *p* < 0.01.

**Figure 5 molecules-28-00791-f005:**
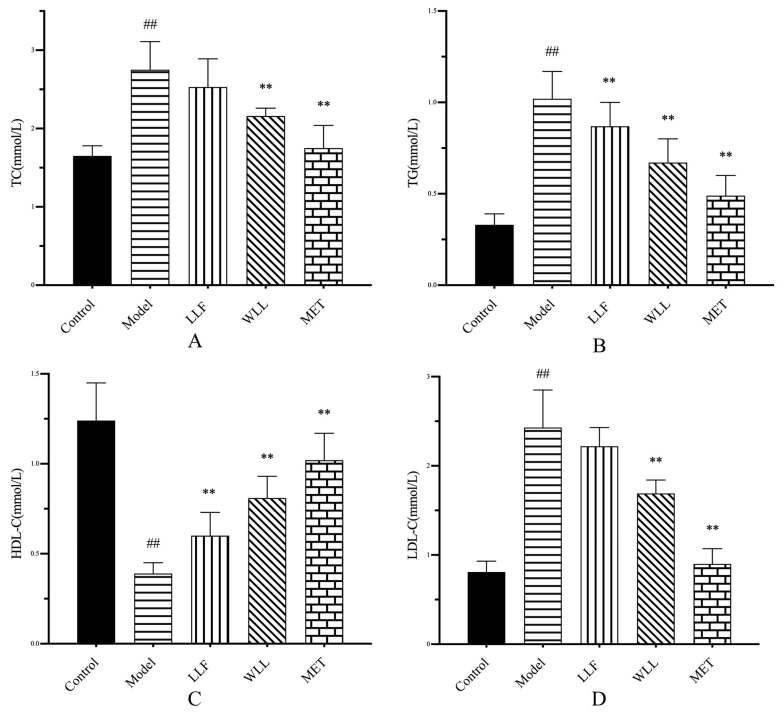
Effects of LLF and WLL extracts on (**A**) TC, (**B**) TG, (**C**) HDL-C, and (**D**) LDL-C in DN rats. Significant differences with the Control group were designated as ^##^
*p* < 0.01. Significant differences with the Model group were designated as ** *p* < 0.01.

**Figure 6 molecules-28-00791-f006:**
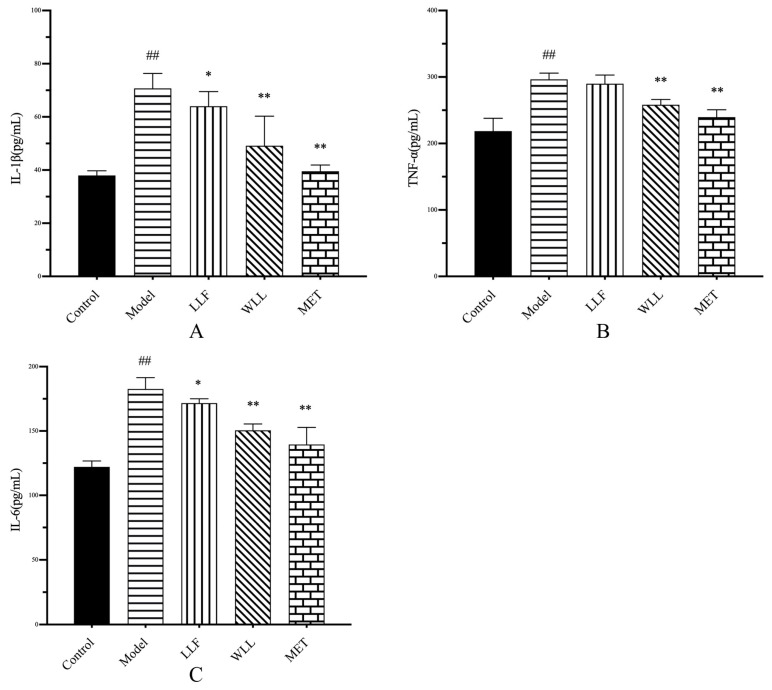
Effects of LLF and WLL extracts on (**A**) IL-1β, (**B**) TNF-α, (**C**) IL-6 in DN rats. Significant differences with the Control group were designated as ^##^
*p* < 0.01. Significant differences with the Model group were designated as * *p* < 0.05, ** *p* < 0.01.

**Figure 7 molecules-28-00791-f007:**
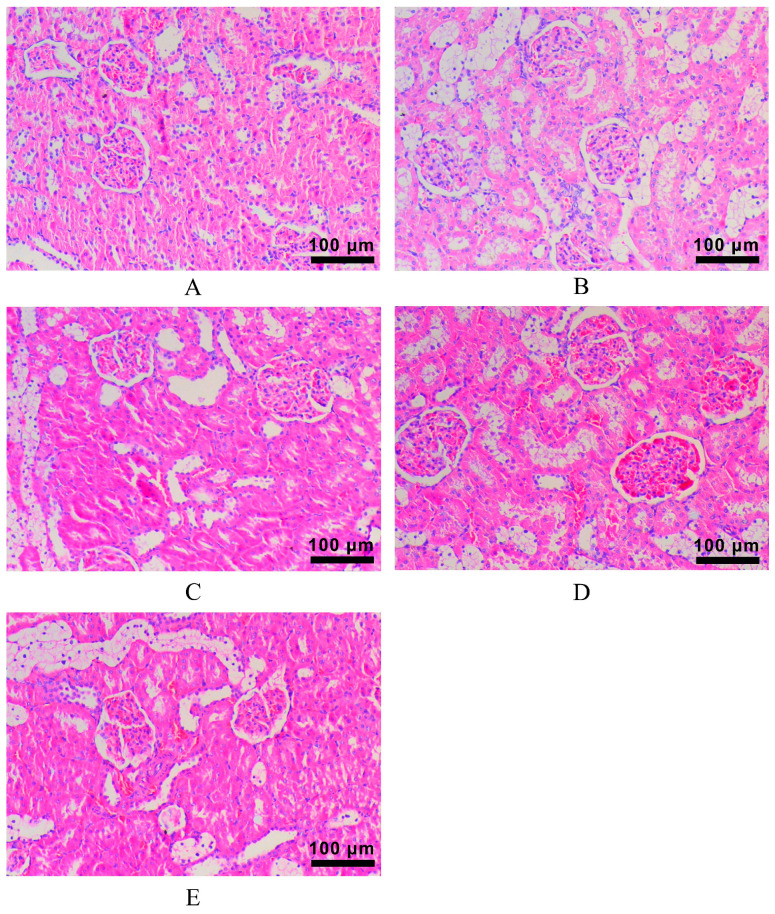
The effect of LLF and WLL extracts on kidney histopathological changes in DN rats (×200), stained with HE. (**A**) Control group kidney tissue; (**B**) Model group kidney tissue; (**C**) LLF group kidney tissue; (**D**) WLL group kidney tissue; (**E**) MET group kidney tissue.

**Figure 8 molecules-28-00791-f008:**
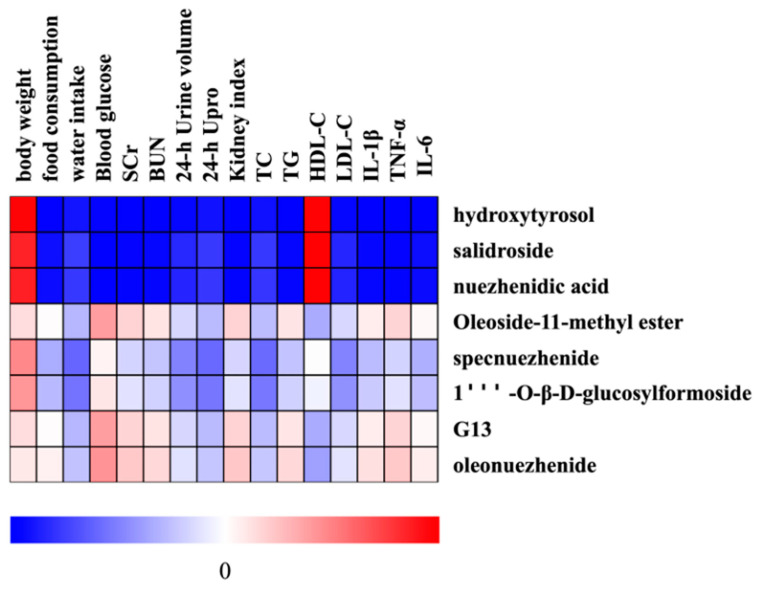
Heatmap of the correlations between the active components of LLF and WLL extracts and physicochemical parameters of kidney protective effects in rats.

**Figure 9 molecules-28-00791-f009:**
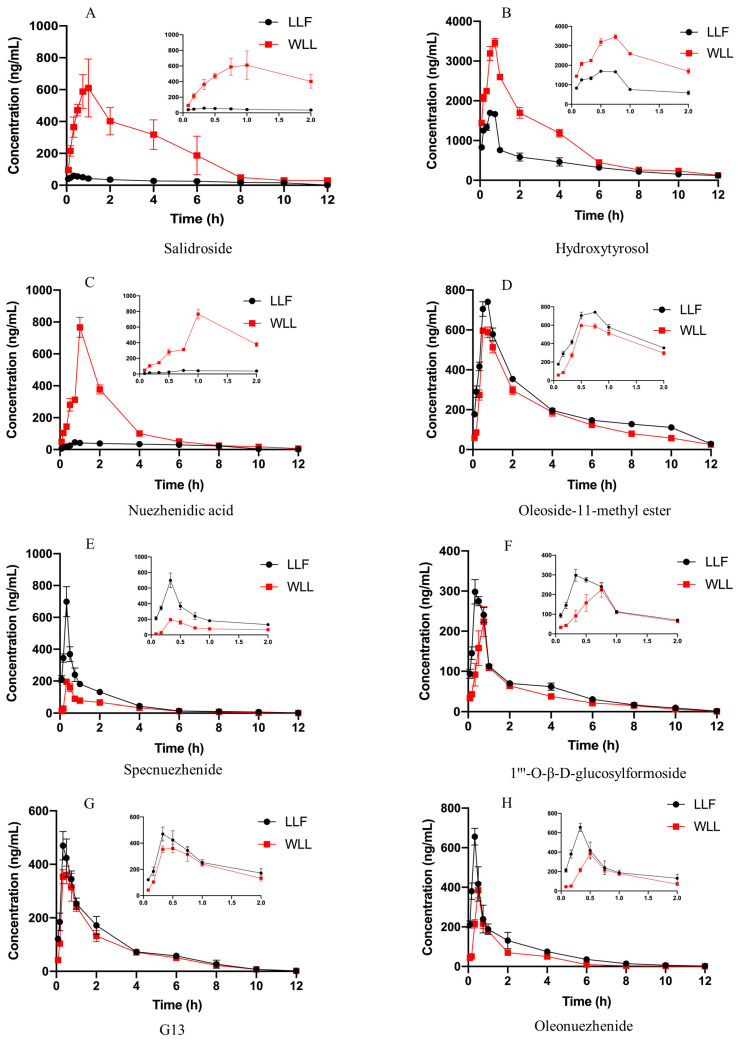
Mean plasma concentration-time curves of salidroside (**A**), hydroxytyrosol (**B**), nuezhenidic acid (**C**), oleoside-11-methyl ester (**D**), specnuezhenide (**E**), 1‴-O-β-d-glucosylformoside (**F**), G13 (**G**), and oleonuezhenide (**H**) in DN rats after oral administration of LLF and WLL extracts.

**Figure 10 molecules-28-00791-f010:**
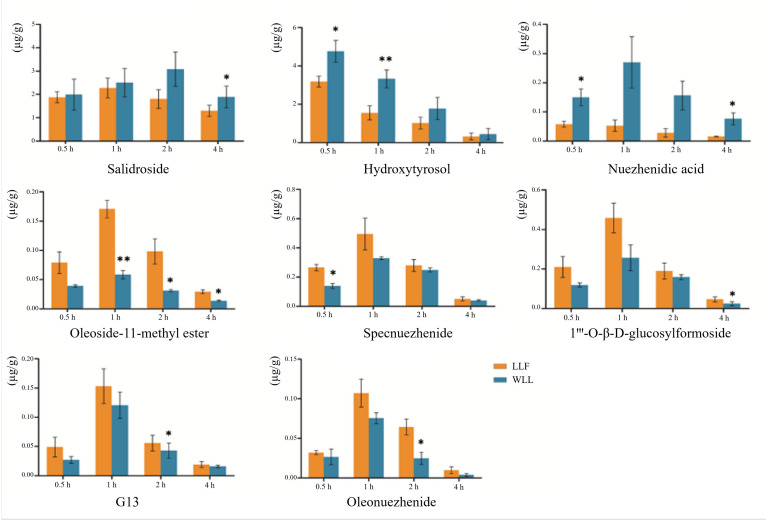
Kidney distribution profiles of the eight components in rats after oral administration of LLF and WLL extracts. Values are presented as the mean ± SD, *n* = 6. Significant differences with LLF group were designated as * *p* < 0.05, ** *p* < 0.01.

**Figure 11 molecules-28-00791-f011:**
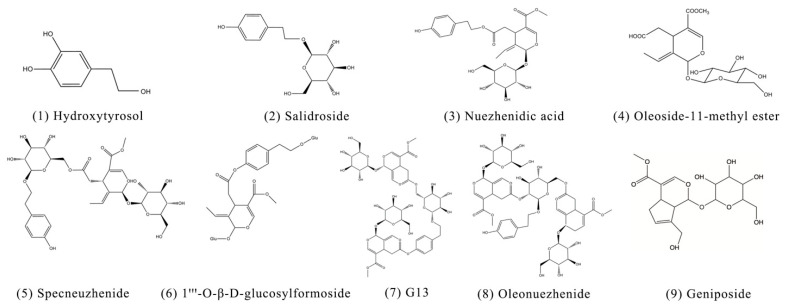
Chemical structures of (**1**) hydroxytyrosol, (**2**) salidroside, (**3**) nuezhenidic acid, (**4**) oleoside-11-methyl ester, (**5**) specnuezhenide, (**6**) 1‴-O-β-d-glucosylformoside, (**7**) G13, (**8**) oleonuezhenide, and (**9**) geniposide.

**Table 1 molecules-28-00791-t001:** Calibration curve, r, and linear range for the eight components (*n* = 5).

Components	Calibration Curves	r	Linear Range (µg)
Hydroxytyrosol	Y = 970,807X − 3559.8	0.9999	0.001–0.2
Salidroside	Y = 113,443X + 685.48	0.9997	0.009–1.8
Nuezhenidic acid	Y = 649,376X − 6312.8	0.9998	0.01–2.0
Oleoside-11-methyl ester	Y = 490,125X + 212	0.9997	0.001–0.2
Specnuezhenide	Y = 549,540X − 12,082	0.9998	0.019–3.8
1‴-O-β-d-glucosylformoside	Y = 489,602X + 2275.4	0.9997	0.001–0.2
G13	Y = 617,621X – 37,494	0.9994	0.01–2.0
Oleonuezhenide	Y = 498,752X − 158.75	0.9999	0.008–1.6

**Table 2 molecules-28-00791-t002:** Contents of the eight active ingredients in extracts of LLF and WLL (mean ± SD, *n* = 5, mg/g).

Ingredients	LLF Extract	WLL Extract
Hydroxytyrosol	2.70 ± 0.35	9.23 ± 0.54
Salidroside	6.93 ± 0.52	274.35 ± 0.57
Nuezhenidic acid	1.75 ± 0.63	36.56 ± 0.58
Oleoside-11-methyl ester	42.66 ± 0.53	3.92 ± 0.87
Specnuezhenide	252.01 ± 0.61	106.48 ± 0.48
1‴-O-β-d-glucosylformoside	6.23 ± 0.35	2.32 ± 0.42
G13	148.84 ± 0.68	13.82 ± 0.58
Oleonuezhenide	103.28 ± 0.36	4.35 ± 0.478

**Table 3 molecules-28-00791-t003:** The body weight, food consumption, and water intake (g) (*n* = 10).

Group	Body Weight (g)	Food Consumption (g/24 h)	Water Intake (mL/24 h)
Control	542.18 ± 39.27	24.82 ± 0.84	49.55 ± 6.64
Model	275.35 ± 49.32 ^##^	48.26 ± 1.86 ^##^	340.00 ± 15.12 ^##^
LLF	297.64 ± 48.47	45.54 ± 1.95 *	228.00 ± 17.50 **
WLL	377.91 ± 51.10 **	43.28 ± 3.33 **	170.00 ± 11.64 **
MET	385.34 ± 42.77 **	39.46 ± 2.51 **	200.00 ± 12.12 **

Significant differences with control group were designated as ^##^
*p* < 0.01. Significant differences with Model group were designated as * *p* < 0.05, ** *p* < 0.01.

**Table 4 molecules-28-00791-t004:** Calibration curve, r, linear range, and LLOQ for the eight analytes in the plasma (*n* = 6).

Components	Calibration Curves	r	Linear Range (ng/mL)	LLOQ (ng/mL)
Salidroside	Y = 0.4667X − 0.6966	0.9995	0.5–1000	0.5
Hydroxytyrosol	Y = 0.773X + 0.6044	0.9999	1.8–3600	1.8
Nuezhenidic acid	Y = 2.224X + 8.482	0.9995	0.5–1000	0.5
Oleoside-11-methyl ester	Y = 0.7759X − 0.897	0.9991	0.5–1000	0.5
1‴-O-β-D-glucosylformoside	Y = 3.254X + 26.363	0.9995	0.5–1000	0.5
Specnuezhenide	Y = 10.039X + 96.865	0.9994	0.5–1000	0.5
G13	Y = 0.152X + 5.2572	0.9999	0.3–600	0.3
Oleonuezhenide	Y = 0.4438X + 2.6225	0.9994	0.3–600	0.3

**Table 5 molecules-28-00791-t005:** Pharmacokinetic parameters of the eight ingredients after oral administration of LLF and WLL extracts (*n* = 6, mean ± SD).

Components	Parameters	LLF	WLL
Salidroside	AUC_0–12h_ (ng/mL·h)	297.03 ± 40.54	2527.81 ± 489.80 **
	AUC_0–∞_ (ng/mL·h)	423.82 ± 142.2	2570.35 ± 492.79 **
	MRT_0–12_ (h)	4.27 ± 0.38	3.24 ± 0.41 **
	MRT_0–∞_ (h)	3.81 ± 0.17	3.70 ± 0.39
	t_1/2z_ (h)	6.57 ± 2.79	2.06 ± 0.61 **
	T_max_ (h)	0.39 ± 0.10	0.90 ± 0.14 **
	CLz/F (L/h/kg)	0.04 ± 0.03	0.01 ± 0.00 **
	Vz/F (L/kg)	0.37 ± 0.28	0.02 ± 0.01 **
	C_max_ (mg/L)	60.06 ± 1.24	697.01 ± 131.21 **
Hydroxytyrosol	AUC_0–12h_ (ng/mL·h)	5021.97 ± 474.75	10,902.26 ± 338.71 **
	AUC_0–∞_ (ng/mL·h)	5580.80 ± 387.88	11,473.66 ± 426.50 **
	MRT_0–12_ (h)	3.64 ± 0.07	3.04 ± 0.02 **
	MRT_0–∞_ (h)	5.29 ± 0.15	3.70 ± 0.23 **
	t_1/2z_ (h)	3.75 ± 0.57	3.06 ± 0.72
	T_max_ (h)	0.50 ± 0.00	0.75 ± 0.00
	CLz/F (L/h/kg)	3.60 ± 0.26	1.31 ± 0.05 **
	Vz/F (L/kg)	19.59 ± 4.48	5.75 ± 1.21
	C_max_ (mg/L)	1694.35 ± 5.23	3462.78 ± 108.62 **
Nuezhenidic acid	AUC_0–12h_ (ng/mL·h)	286.53 ± 21.71	1615.14 ± 53.47 **
	AUC_0–∞_ (ng/mL·h)	288.48 ± 22.03	1639.94 ± 64.38 **
	MRT_0–12_ (h)	4.30 ± 0.02	2.43 ± 0.06 **
	MRT_0–∞_ (h)	4.37 ± 0.02	2.58 ± 0.05 **
	t_1/2z_ (h)	1.17 ± 0.41	2.24 ± 0. 60 *
	T_max_ (h)	0.75 ± 0.00	1.00 ± 0.00
	CLz/F (L/h/kg)	0.05 ± 0.00	0.01 ± 0.00
	Vz/F (L/kg)	0.09 ± 0.03	0.03 ± 0.01 **
	C_max_ (mg/L)	45.50 ± 3.00	766.20 ± 62.39 **
Oleoside-11-methyl ester	AUC_0–12h_ (ng/mL·h)	2537.42 ± 67.83	2021.07 ± 99.67 *
	AUC_0–∞_ (ng/mL·h)	4201.31 ± 1029.35	2157.23 ± 179.31 **
	MRT_0–12_ (h)	3.78 ± 0.02	3.51 ± 0.09
	MRT_0–∞_ (h)	5.71 ± 1.63	4.16 ± 0.41
	t_1/2z_ (h)	11.51 ± 5.78	2.98 ± 0.83
	T_max_ (h)	0.67 ± 0.14	0.58 ± 0.14
	CLz/F (L/h/kg)	3.71 ± 0.87	6.99 ± 0.56
	Vz/F (L/kg)	56.90 ± 14.56	29.67 ± 6.61 **
	C_max_ (mg/L)	744.81 ± 9.29	605.62 ± 14.86 **
Specnuezhenide	AUC_0–12h_ (ng/mL·h)	774.79 ± 38.60	348.81 ± 7.30 **
	AUC_0–∞_ (ng/mL·h)	779.95 ± 43.36	349.12 ± 7.50 **
	MRT_0–12_ (h)	1.91 ± 0.10	2.43 ± 0.05 **
	MRT_0–∞_ (h)	2.00 ± 0.20	2.44 ± 0.04 *
	t_1/2z_ (h)	1.71 ± 0.47	0.95 ± 0.26
	T_max_ (h)	0.33 ± 0.00	0.33 ± 0.00
	CLz/F (L/h/kg)	25.70 ± 1.50	57.30 ± 1.23 **
	Vz/F (L/kg)	62.88 ± 13.77	78.34 ± 20.06
	C_max_ (mg/L)	698.99 ± 94.38	195.09 ± 6.81 *
1‴-O-β-d-glucosylformoside	AUC_0–12h_ (ng/mL·h)	901.37 ± 92.32	531.78 ± 54.16 *
	AUC_0–∞_ (ng/mL·h)	908.80 ± 85.36	534.65 ± 53.68 *
	MRT_0–12_ (h)	2.30 ± 0.05	2.00 ± 0.05 *
	MRT_0–∞_ (h)	2.40 ± 0.17	2.07 ± 0.07
	t_1/2z_ (h)	1.66 ± 0.60	1.74 ± 0.26
	T_max_ (h)	0.33 ± 0.00	0.50 ± 0.00
	CLz/F (L/h/kg)	22.14 ± 2.18	37.65 ± 3.62 *
	Vz/F (L/kg)	54.45 ± 25.39	94.25 ± 17.23
	C_max_ (mg/L)	655.39 ± 42.24	384.75 ± 38.68 **
G13	AUC_0–12h_ (ng/mL·h)	1025.82 ± 77.41	881.95 ± 39.80 *
	AUC_0–∞_ (ng/mL·h)	1072.76 ± 115.48	894.63 ± 38.36 *
	MRT_0–12_ (h)	2.54 ± 0.26	2.69 ± 0.16
	MRT_0–∞_ (h)	2.73 ± 0.38	2.85 ± 0.20
	t_1/2z_ (h)	1.96 ± 1.08	1.80 ± 0.25
	T_max_ (h)	0.39 ± 0.10	0.44 ± 0.10
	CLz/F (L/h/kg)	14.094 ± 1.56	16.79 ± 0.71
	Vz/F (L/kg)	38.91 ± 18.38	43.69 ± 7.59
	C_max_ (mg/L)	500.50 ± 20.10	377.69 ± 16.56 *
Oleonuezhenide	AUC_0–12h_ (ng/mL·h)	605.12 ± 26.46	437.77 ± 9.87 *
	AUC_0–∞_ (ng/mL·h)	620.13 ± 27.33	446.85 ± 16.81 *
	MRT_0–12_ (h)	2.80 ± 0.06	2.86 ± 0.10
	MRT_0–∞_ (h)	2.84 ± 0.12	2.89 ± 0.08
	t_1/2z_ (h)	2.11 ± 0.57	2.05 ± 0.56
	T_max_ (h)	0.39 ± 0.10	0.75 ± 0.00
	CLz/F (L/h/kg)	24.22 ± 1.05	33.60 ± 1.25 *
	Vz/F (L/kg)	73.57 ± 19.76	98.52 ± 23.73
	C_max_ (mg/L)	300.54 ± 27.04	223.92 ± 37.50

Note: * *p* < 0.05, ** *p* < 0.01.

**Table 6 molecules-28-00791-t006:** Calibration curve, r, linear range, and LLOQ for eight analytes in kidney tissue (*n* = 6).

Components	Calibration Curves	r	Linear Range (ng/mL)	LLOQ (ng/mL)
Salidroside	Y = 0.0018X + 0.0011	9.00–18,000	0.9998	9.00
Hydroxytyrosol	Y= 0.4253X − 49.7400	12.50–25,000	0.9998	1.50
Nuezhenidic acid	Y = 0.5417X − 15.042	2.75–5500	0.9996	2.75
Oleoside-11-methyl ester	Y = 0.1266X + 0.0019	1.50–3000	0.9998	1.50
1‴-O-β-d-glucosylformoside	Y= 0.4822X + 24.4800	3.50–7000	0.9991	3.50
Specnuezhenide	Y = 1.4842X + 30.094	12.50–25,000	0.9993	12.50
G13	Y = 0.2276X − 4.5269	0.50–1000	0.9998	0.50
Oleonuezhenide	Y = 0.2761X + 3.5803	0.50–1000	0.9993	0.50

**Table 7 molecules-28-00791-t007:** Kidney distribution profiles of the eight ingredients after oral administration of LLF and WLL extracts (*n* = 6, mean ± SD).

Components	Extracts	Concentration (Mean ± SD, µg/g)			
		30 min	60 min	120 min	240 min
Salidroside	LLF	1.876 ± 0.234	2.280 ± 0.432	1.805 ± 0.402	1.299 ± 0.237
	WLL	1.995 ± 0.665	2.506 ± 0.602	3.080 ± 0.738	1.894 ± 0.469 *
Hydroxytyrosol	LLF	3.191 ± 0.282	1.551 ± 0.375	1.016 ± 0.314	0.321 ± 0.177
	WLL	4.763 ± 0.568 *	3.334 ± 0.464 **	1.777 ±0.584	0.442 ± 0.288
Nuezhenidic acid	LLF	0.057 ± 0.010	0.053 ± 0.019	0.028 ± 0.015	0.016 ± 0.001
	WLL	0.150 ± 0.028 *	0.270 ± 0.088	0.157 ± 0.050	0.076 ± 0.020 *
Oleoside-11-methyl ester	LLF	0.079 ± 0.019	0.171 ± 0.015	0.099 ± 0.021	0.029 ± 0.003
	WLL	0.039 ± 0.002	0.059 ± 0.007 **	0.031 ± 0.002 *	0.014 ± 0.001 *
1‴-O-β-d-glucosylformoside	LLF	0.210 ± 0.053	0.458 ± 0.075	0.190 ± 0.040	0.047 ± 0.013
	WLL	0.119 ± 0.011	0.256 ± 0.066	0.159 ± 0.013	0.024 ± 0.011 *
Specnuezhenide	LLF	0.266 ± 0.022	0.494 ± 0.109	0.280 ± 0.040	0.050 ± 0.016
	WLL	0.139 ± 0.017 *	0.330 ± 0.010	0.250 ± 0.014	0.040 ± 0.005
G13	LLF	0.049 ± 0.017	0.153 ± 0.030	0.056 ± 0.013	0.019 ± 0.005
	WLL	0.027 ± 0.006	0.121 ± 0.022	0.043 ± 0.013 *	0.016 ± 0.002
Ole	LLF	0.032 ± 0.002	0.107 ± 0.018	0.064 ± 0.010	0.010 ± 0.004
	WLL	0.026 ± 0.010	0.075 ± 0.007	0.025 ± 0.008 *	0.004 ± 0.001

Note: Significant differences with LLF group were designated as * *p* < 0.05, ** *p* < 0.01.

## Data Availability

The data used to support the findings of this study are available from the corresponding author upon request.

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
