# Peer review of "Pharmacodynamics, Pharmacokinetics, and Kidney Distribution of Raw and Wine-Steamed Ligustri Lucidi Fructus Extracts in Diabetic Nephropathy Rats"

_molecules, 2023, doi:10.3390/molecules28020791_

Round 1

Reviewer 1 Report

The purpose of this study was to investigate the differences of pharmacodynamic, phar- 12 macokinetic and kidney distribution between Ligustri Lucidi Fructus (LLF) and wine-steamed 13 Ligustri Lucidi Fructus (WLL) extracts in diabetic nephropathy (DN) rats. The structure of the manuscript is well designed and parameters used for the very appropriate for the study. The story is interesting and data seems solid enough to support the conclusion. In my view as a reviewer, I recommend revise, and there are some specific questions should be answered as follow.

1. It is suggested to add the pharmacological effect of wine steamed Ligustri Lucidi Fructus.

2. Authors are suggested to provide clinical applications of the wine steamed Ligustri Lucidi Fructus extracts in diabetic nephropathy in human subject.

3. Some nouns in the text should be unified and case sensitive, for example “traditional Chinese medicine”.

4. why choose the eight ingredients to carry out the pharmacokinetic and kidney distribution study, add discussion.

5. The discussion part need to further the results part. Modified the discussion to better explain the value of the research.

6.There are some language issues specifically in result section and legends of figures which should be checked.

Reviewer 2 Report

1. The authors have done a good work with very good data related to pharmacodynamics, pharmacokinetics and kidney distribution of Raw and wine-steamed Ligustri Lucidi Fructus Extracts in diabetic nephropathy rats. However, the authors have to justify their work on HPLC.

2. They should provide justification for using HPLC rather than LCMS as it has good sensitivity and specificity.

3. Please discuss the pharmacodynamic and pharmacokinetic correlation.

Reviewer 3 Report

Please see attached pdf file.
